# Recent Environmental Legislation in Brazil and the Impact on Cerrado Deforestation Rates

Carlos Henrique Pires Luiz *  and Valdir Adilson Steinke 

Graduate Program in Geography, Department of Geography, University of Brasília, Brasília 70.910-900, Brazil; valdirsteinke@gmail.com

*   Correspondence: cpiresluiz@gmail.com

**Abstract:** This study aims to relate the recent trajectory of Brazilian environmental policies with the last 20 years deforestation rates observed in the Cerrado through the PRODES-Cerrado monitoring initiative. The main hypothesis is that the improvement of environmental legislation in Brazil, mainly during the period between 2005 and 2012, influenced the decrease in deforestation rates. In addition, policies to control environmental compliance, such as the Rural Environmental Registry (CAR) also influenced the reduction of deforestation. In the early 2000s, there was a significant increase in public environmental policies and implementation of an environmental management structure with the creation of conservation, protection, and management agency for conservation units (ICMBio), the Brazilian Forest Service for the management of public forests and Rural Environment Registry (CAR). Comparing the annual deforestation rate, it was observed that between 2000 and 2005, when 12.60% of the Cerrado was deforested, the annual deforestation average rate was 2.52%. Between 2006 and 2012, the period that precedes the revision of the Forest Code, the total deforestation is equivalent to 7.98%, which results in an annual average rate of 1.33%. After the enactment of the new Forest Code, between 2013 and 2020, there was a deforestation of 7.03% of the Cerrado area, which is equivalent to a deforestation annual average rate of 1.00%. One of the positive aspects of the new legislation was the creation of CAR, which obliges rural landowners to make an environmental attributes declaration of their property, this program being the main environmental management tool created in Brazil recently. After CAR regulation in 2014, there was a decrease in deforestation from 10,904 $km^2$ to 7905 $km^2$ in 2020. On the other hand, since 2016, changes occurred in the political scenario that increased agribusiness influence and the rise of a more conservative agenda, which jeopardizes the future of environmental quality in Brazil (illustrated, for example, by the increased release of pesticides from 104 in 2010 to 493 in 2020). As the main conclusion of this research, we showed that the state's commitment to environmental management can contribute to deforestation reduction. The regulation of programs such as CAR can also contribute to the reduction of deforestation since it is one more tool for monitoring and ensure compliance of environmental regularization and recovery vegetation programs. At the same time, is necessary to keep on monitoring deforestation once the influence of the agricultural lobby has gained strength.

**Keywords:** Cerrado; deforestation; CAR; PRODES Cerrado; Brazil environmental policy

## 1. Introduction

The Cerrado formations in Brazil cover about 22% of the territorial extension of the country, covering 12 states and housing strategic watersheds such as São Francisco and Tocantins-Araguaia [1]. According to [2], about 43% of the Cerrado's native vegetation cover has been suppressed since the Brazil colonial period, giving rise to urban centers, agricultural surfaces, mining areas, and reforestation with non-native species. The Cerrado is the second most devastating biome in Brazil, undergoing less modification only than the Atlantic Forest [3,4]. The changes in land cover are part of a set of transformations that

began in the colonial period, whose potential for interference has been expanded over time (mainly since this Biome concentrates the majority of inhabitants in Brazil).

While much attention has been placed on the Brazilian Amazon, the native vegetation suppression in Cerrado is higher [5]. Studies such [6,7], found that the causes of the increase in deforestation, is related to the growth of livestock, increase in the number of highways and effectiveness of conservation policies, such as the fact that Cerrado has five times less protected areas than the Amazon.

Before this scenario, concerns about the advance of deforestation have assumed a prominent role in research and also in governance actions of Brazil's environmental agenda. Changes in government and the direction of environmental policies in recent times prove the importance of this concern, especially with the increase in deforestation rates. The composition of the Brazilian Congress since 2014 has been dominated by a powerful lobby of the ruralist contingent and the political instability that Brazil faces in recent years threatens the environmental progress, achieved especially in the 2000s [8].

The relation between policy and deforestation, had been calling attention of many studies, such [9], that related deforestation to the recent weakening of the Ministry of the Environment's deforestation enforcement actions. Ref. [10], has associated the recent burning crisis in Amazon with deforestation and the fact that while government's claims that the Amazon fire situation in August 2019 was "normal", the deforestation in 2019 was almost four times the average from 2016–2018. Ref. [11], published a document denouncing that the current Brazilian political and socio-environmental scenario demonstrates the result of the dismantling carried out by the actual government, the constant attacks against socio-environmental bodies and entities, and speeches against the performance of servers and environmental standards.

Since the beginning of the government's agenda, they observed an increase in the number and extent of forest fires, expansion of deforestation, and decrease in field inspection actions [11].

Although discussion of changes in legislation is pertinent, the exhaustive comparison of the Forest Code versions since its first promulgation in 1934 does not necessarily effectively contribute to the reflection on environmental impacts, bearing in mind that while earlier versions were more restrictive for Cerrado areas, its compliance was not effective [12]. This lack of compliance with environmental legislation in previous versions of the Forest Code is related to the development of mechanisms for monitoring.

The Brazilian Forest Code of 2012 [13] has been in effect since April 2012 and environmental changes and amnesties, addressed in the studies by [14–18], show the need to improve the effectiveness of law enforcement through actions such as increasing monitoring, inspection, and effectiveness of administrative and/or criminal penalties for infringers.

In this context, the Environmental Rural Registry or 'CAR' (Cadastro Ambiental Rural, in Portuguese) is the main tool in environmental management in Brazil, where the owners of rural properties make the declaration of the environmental attributes of their property.

The goal of this CAR declaration is to provide a way to enable the land houses to achieve compliance with environmental legislation. After this declaration, the validation of the information provided is a role of the environmental state secretariats. While this declaration is not validated by the government, it can be a problem since sometimes the information does not fit with the real environmental compliance of the property. Within the scope of the CAR, the information that must be declared by the landowners is the boundary of the property, the vegetation coverage, the permanent preservation areas (APP), areas with restricted use, and legal reserve areas (RL).

According to data from the Brazilian Forest Service [19] until December 2020 there were just over 6.9 million rural properties registered across the country, and in the Cerrado this number is almost 1 million properties. Adherence numbers to the CAR, a program regulated in 2014, exceeded the initial expectations of the registrable area by just under 20%, with 397.8 million ha expected and currently there is already 489.2 million ha. Given

these numbers, the CAR has the potential to be used as a governance tool since its database concentrates so much environmental information.

The regulation of CAR through Normative Ruling No. 2 from Ministry of Environment (MMA) of 6 May 2014 [20], established an initial term of adhesion of one year, so the landowners would have until 5 May 2015, for registration in CAR. However, the deadline has been postponed five times. This procrastination reflects the growing influence of agribusiness on environmental policy, the low initial adherence to the program and also is related to environmental behaviors, such as: motivational, moral, context and habit [21]. In this case, the motivation for CAR registration is predominantly normative [22] and not necessarily due to a belief in the importance of environmental preservation [23].

Given this scenario of relatively recent changes in legislation with the implementation of the 2012 Forest Code and the instrumentalization of the CAR, is it possible to trace any relationship between the deforestation rates of the Cerrado and the most recent legal instruments in Brazil? Is it possible to find any relationship between the behavior of the country's environmental policy and deforestation rates? To answer these questions, the present study aims to develop a comparison between the deforestation rates mapped by PRODES Cerrado from 2000 to 2020 [24], the main frameworks of environmental legislation and the adherence incremented in CAR.

## 2. Methodology

The methodological steps followed, showed in Figure 1, started with a review of the main milestones of environmental legislation from 2000 to 2020. Then, based on deforestation data from Cerrado [24], the main regulatory frameworks were superimposed on the timeline along with the trajectory of deforestation. The objective of this approach is to study, in an exploratory way, how the stronger presence of the State, through the creation of laws and regulatory agencies, can influence the decline in deforestation rates.

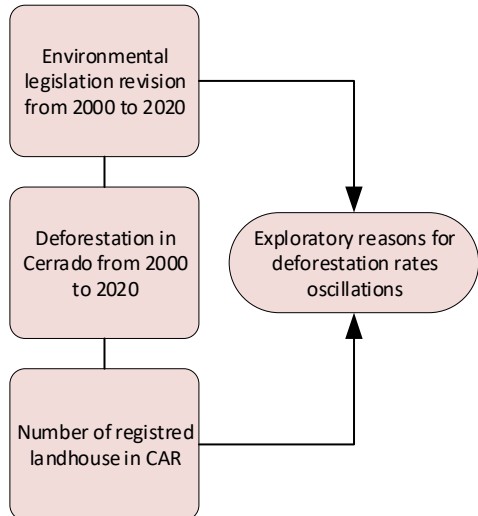

**Figure 1.** Methodology steps.

As the CAR is the main public policy to control environmental compliance in Brazil [25], also in an exploratory way, a correlation analysis was proposed between the increasing number of land houses registered in CAR over the years and the rates of deforestation from PRODES.

The intervals chosen to present some results are related to important milestones for Brazilian environmental policy: 2012–2020, the impact of Brazilian Forest Code change; 2005–2012, the impact of the Cerrado Biome Conservation and Sustainability Program; 2000–2005, period with available deforestation data and before an increase in the implementation of environmental policies.

The database used in this research is composed of deforestation data obtained from PRODES [24], which is the official basis for monitoring deforestation in Brazil. Following the same methodology as PRODES Amazônia, PRODES Cerrado [24] uses TM/Landsat5, ETM+/Landsat7, OLI/Landsat8 and LISS-III/RESOURCESAT2 images to calculate the rate of deforestation from 2000 [26]. We decided to use PRODES deforestation data, since it has the largest monitoring data information available of deforestation in Cerrado from 2000 to 2020, and this information is used by public agencies such as Ministry of the Environment, Ministry of Agriculture, Brazilian Forestry Service, Chico Mendes Institute for the Biodiversity Conservation and Brazilian Institute for the Environment and Natural Resources for public policies decision.

To carry out the assessment of deforestation inside and outside protected areas, data were collected from ICMBio [27], which is the institution responsible for managing protected areas in Brazil.

To contextualize the political-environmental scenario of the last 20 years, a search was carried out in the environmental legislation for the main political landmarks that is presented in Section 3.

The data recorded in the CAR were used in an exploratory way to identify any correlation in deforestation rates and adherence to this public policy.

## 3. The Trajectory of Environmental Legislation in Brazil since the Year 2000 and the Environmental Political Context

To further relate the Cerrado deforestation rates with the main legislative and contextual frameworks associated with the environment, it is necessary to construct the trajectory of these political frameworks from 2000, period which initiated the monitoring deforestation by INPE in the Cerrado. In Figure 2 a timeline is presented with the main political frameworks of the period and the amount of observed deforestation in Cerrado.

In 2000, the Law 9985/2000 [28] established the National Conservation Units System (SNUC). This law regulated the categories of protected areas (PA), and established criteria and norms for the creation, implementation and management of protected areas. In the Cerrado Biome there are currently 133 PA of the integral protected category and 158 of the sustainable use, as illustrated in Figure 3.

The Integral protected conservation unit category prevents areas from deforestation, based on use restriction. Together these areas protect the equivalent of 49,934 km$^2$, which is about 2.45% of the entire area of the Cerrado, considering that this Biome has 2,036,448 km$^2$. Comparing to Amazon biome, where the protected areas proportion cover 10.20% of biome, Cerrado has few areas for protection which represents a risk for a global biodiversity hotspot [4]. In the Sustainable Protected conservation units, there are fewer restrictions concerning the conversion of the forest and are allowed the presence of the traditional population and some land-use activities, such as extractivism. Consequently, the amount of deforestation (13,550 km$^2$ until 2020) inside that area is 23 times bigger than inside the Restricted Protected conservation units (589 km$^2$) [24].

In 2002, there was the Rio+10 Conference, whose main objective was to monitor compliance with the actions planned at Eco-92 (United Nations Conference on Environment and Development). Besides, it discussed social and quality of life aspects such as sustainable development, poverty, water use, renewable energy sources and management of natural resources. This conference also focused on climate change—especially the Clean Development Mechanism (MDL) and the Kyoto Protocol. The signatory countries reaffirmed their commitments to the Agenda 21 goals. However, the document did not set deadlines, which gave the conference not effective results.

Also in 2002, the planning phase of Agenda 21 was completed by the Commission for Sustainable Development Policies and National Agenda 21 (CPDS). The construction of this plan involved participative management, with the collaboration of more than 40,000 people [29], whose objective was to expand participatory planning focused on the priorities set for achieving sustainable development. Its implementation began in 2003, and

coincided with the election of President Lula, who established a Multi-annual Action Plan. The document reflected an analysis of the main environmental problems. However, the planning has been running out of steam as a result of new priorities and recent changes in environmental policy [30].

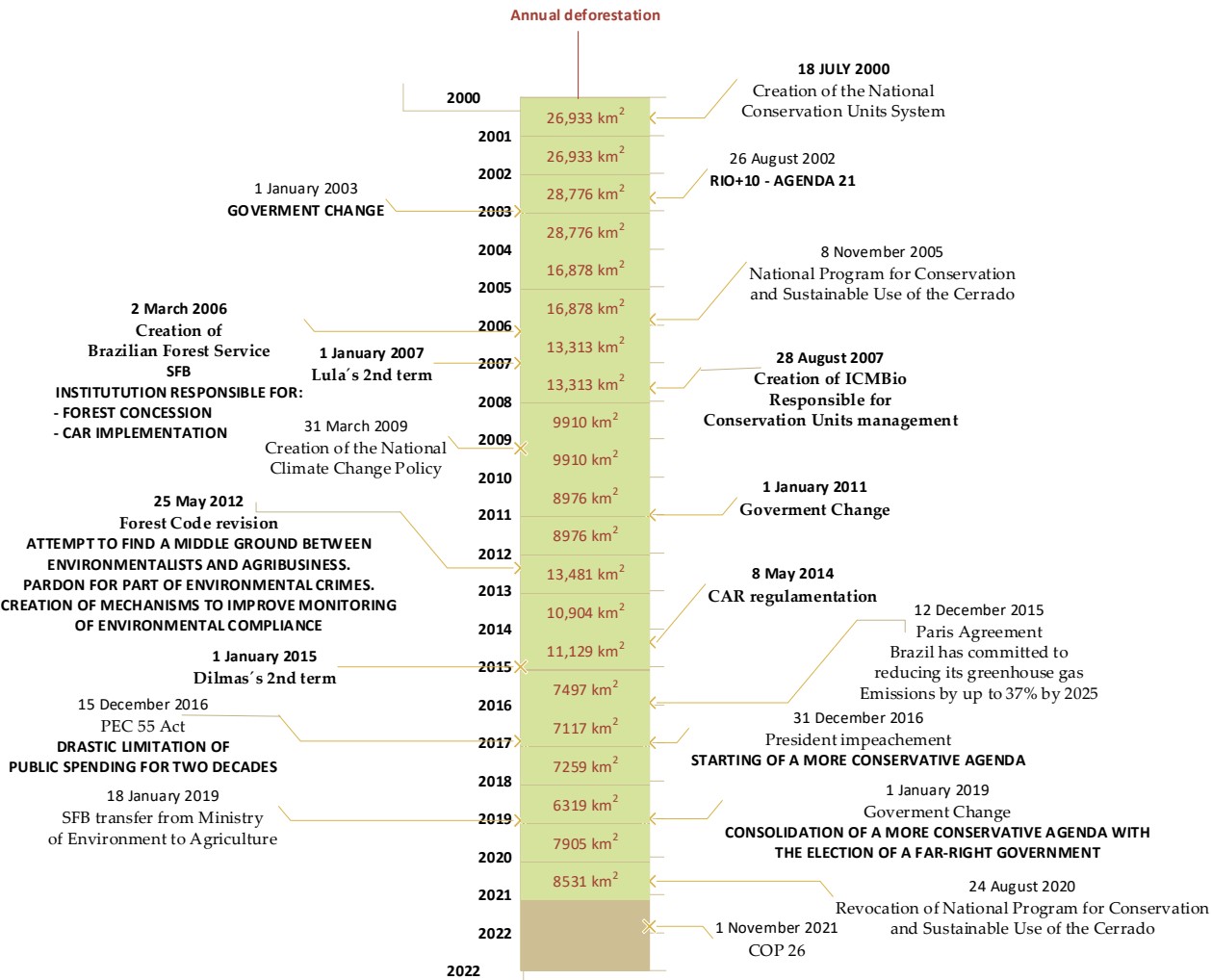

**Figure 2.** Trajectory of environmental legislation in Brazil and deforestation in Cerrado.

By Decree 5577 [31], in 2005, the National Program for Conservation and Sustainable Use of the Cerrado biome was created (i.e., Sustainable Cerrado Program, whose purpose is to promote the conservation, restoration, recovery and sustainable management of the Cerrado biome ecosystems through the appreciation and recognition of their traditional populations). Within this Program, CONACER (the National Commission of the Sustainable Cerrado Program) was created, a collegiate made up of representatives of various government agencies, the main one being the MMA, to monitor the implementation of the Program's actions and other government policies, such as the National Environment Policy, the SNUC, the National Biodiversity Policy, and the National Policy on Climate Changes.

In 2006, through Federal Law 11,284 [32], the legislation of public forest management for sustainable production was established; and establishes, within the structure of the MMA, the Brazilian Forest Service (SFB), an agency that in 2014 was responsible for implementing the CAR; and created the National Forest Development Fund (FNDF). The FNDF's mission is to instigate the development of sustainable forestry activities in Brazil and to promote technological innovation in the sector.

Thereafter, Federal Law 11,428/2006 [33] established criteria for the use and protection of native vegetation in the Atlantic Forest Biome, which indirectly increased pressure on the Cerrado. This law has created more barriers to Atlantic Forest suppression.

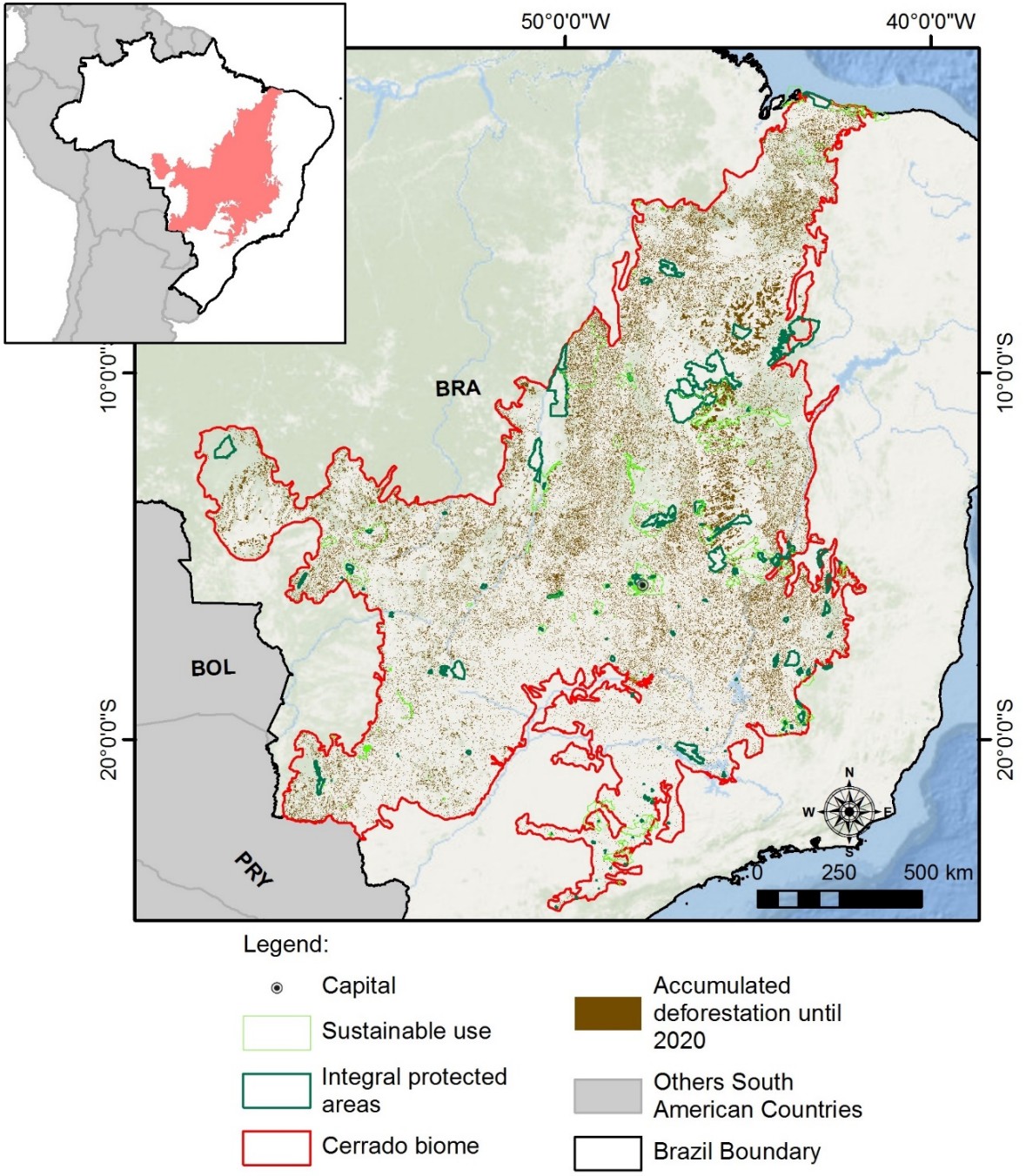

**Figure 3.** Distribution Protection UCs in the Cerrado x the accumulated deforestation area until 2020. Source: prepared by the authors, based on [24,27].

In 2007, by Federal Law 11,516 [34], Chico Mendes Institute for Biodiversity Conservation (ICMBio) was created, linked to the MMA, responsible for the management of SNUC, with the difficult task of overseeing and managing over 800 conservation units. This institute also promotes and runs programs for research, protection, preservation and conservation of biodiversity.

In 2008, the government, through Federal Law 11,828 [35], established the legal taxation applicable to donations received by Brazilian Development Bank (BNDES), and

intended to prevent, monitor and combat deforestation to promote the conservation and sustainable use of Brazilian forests. This law regulates donations received from international funds to combat deforestation.

In 2009, with regard to the consequences of deforestation, special attention was given to the topic of climate change, with the enactment of the laws establishing the National Climate Change Fund (Federal Law 12.114) [36], and the National Climate Change Policy (PNMC) (Law 12.187) [37]. The National Fund for Climate Change aims to raise funds to support projects or studies and financing of ventures aimed at mitigating climate change and adapting to climate change and its effects. The PNMC seeks to establish actions to reconcile economic and social development in line with the protection of the climate system, and its main objective is to reduce greenhouse gas emissions. To achieve these objectives, one of the strategies is to combat deforestation and encourage reforestation.

One of the strategies to combat deforestation is to provide financial compensation to producers without financial resources, so in 2011, Law 12,512 established the Environmental Conservation Support Program and the Rural Productive Activities Promotion Program [38]. These programs aimed to integrate families in extreme poverty who develop conservation activities in areas of national forests, extractive reserves and federal sustainable development reserves, forest settlement projects, sustainable development projects or agro-extractive settlement projects instituted by the national colonization institute for Agrarian Reform (INCRA) and territories occupied by riverside dwellers, extractivists, indigenous, quilombolas, and other traditional communities.

In 2012, the Rio+20 Conference aimed to renew the commitments made at Eco-92 and to evaluate what was carried out after 20 years. Topics such as the real economy, actions to guarantee sustainable development, ways to eradicate poverty and the role of international governance in sustainable development were also discussed. With an economic scenario of recent recovery from the global economic crisis of 2008, the final document of the Conference reflected some intentions, but did not advance the next steps and at practical measures do guarantee environmental protection.

The major change in the Brazilian environmental scenario in recent times was the revision of the CFB, Law 12.651/2012 [13], which established new standards for the protection of the environment, through a policy of encouraging the sustainable use of natural resources. Central themes with APP and RL have undergone some changes, which generated criticism from environmentalists. APP is a permanent preservation area that protects special areas, such as rivers, lakes, steep areas, hilltops, among other characteristics. RL is a legal reserve area, which is a piece of land within a property, that must be preserved. This new version is less strict concerning environmental protection requirements compared to the previous CFB version. The main criticisms are associated with greater alignment with agribusiness interests, culminating in the reduction of APP based on the property size, and amnesty in the recovery of deforested areas within the legal reserve, for lands with an area smaller than 4 fiscal modules, and that were deforested before 2008 [14,39,40]. This resulted in a way of legalizing the irregularities of non-compliance with the previous law [41]. On the other hand, the CFB regulation presents positive aspects such as the creation of CAR, implemented in 2014, a system that allows greater monitoring of violations of environmental legislation and that also introduced payment mechanisms for environmental services such as Environmental Reserve Quotes—CRA and environmental recovery, such as the Environmental Recovery Program—PRA [13].

In 2014, a cooperation program was signed among Brazil, the United Kingdom, and the World Bank to reduce deforestation and burning in the Brazilian Cerrado—Federal Cerrado Project. Initiatives such as the Federal Cerrado Project aim to minimize the adverse effects of climate changes in the Cerrado through environmental management, rural environmental regularization, and forest fire prevention and control. The Project, which is funded by the UK Government's Department of Environment, Food and Rural Affairs (DEFRA) and administered by the International Bank for Reconstruction and Development (IBRD) from

World Bank Group, has received $4.3 million to finance actions divided into components for: prevention and control of deforestation, fire and the promotion of CAR [42].

Still, in 2014, a turning point began in the Brazilian political context, breaking a cycle of economic growth and greater action in environmental causes initiated in 2003.

The trigger for this change was the reelection of President Dilma Roussef with 51.64% of valid votes, this being the closest difference in history. In the face of this, opposing blocs articulated a weakening strategy of the government, which combined with an unstable economic scenario, and the support of the elite, middle class, and press sectors, culminated in the acceptance of the impeachment process in 2016.

From the dismissal of President Dilma Roussef, there has been an increase in the influence of conservative sectors, such as the popularly known ruralist contingent—Parliamentary Front of Agriculture. This sector, which since 2016 holds the majority in the House of Representatives, 225 of the 513 deputies, representing 44% of the total seats and 32 senators of the 81 that make up the Senate, defends the interests of large landowners and policies to stimulate agribusiness, at the same time, are strongly opposed to environmental legislation and land reform.

As a way of illustrating this influence, since 2016, was observed a rise in the amount of the number of type of pesticides approved by the Ministry of Agriculture, increasing 104 in 2010 to 493 in 2020 (Figure 4).

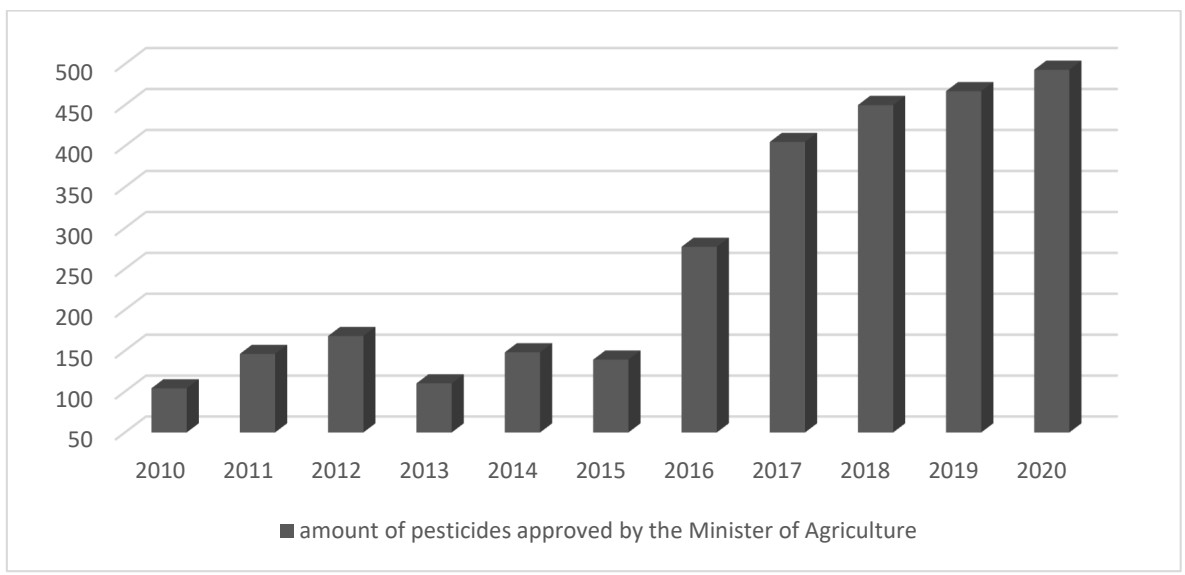

**Figure 4.** Rise in the number of type of pesticides from 2010 to 2020. Source: prepared by the authors, based on [43]. In addition to the growing influence of the ruralist contingent, in 2016, the proposal for Constitutional Amendment (Law Project that modifies the constitution, resulting in specific changes to the constitutional text of Brazil, preserving its immutables clauses.) (PEC) N°. 55 [44] was approved, which established budget limit and spending freeze for 20 years in all sectors except education and health. Thus, we envisage a scenario of investment containment that can increase deforestation rates and put at risk sustainable development and the goals assumed by the country in international agreements.

## 4. PRODES Cerrado and the Rate of Deforestation in the Bioma

Figure 5 shows the evolution of deforestation in the Cerrado in four years: 2000, 2005, 2012, and 2020.

Rates of deforestation have declined since 2015, which may reflect the implementation of CAR in the previous year.

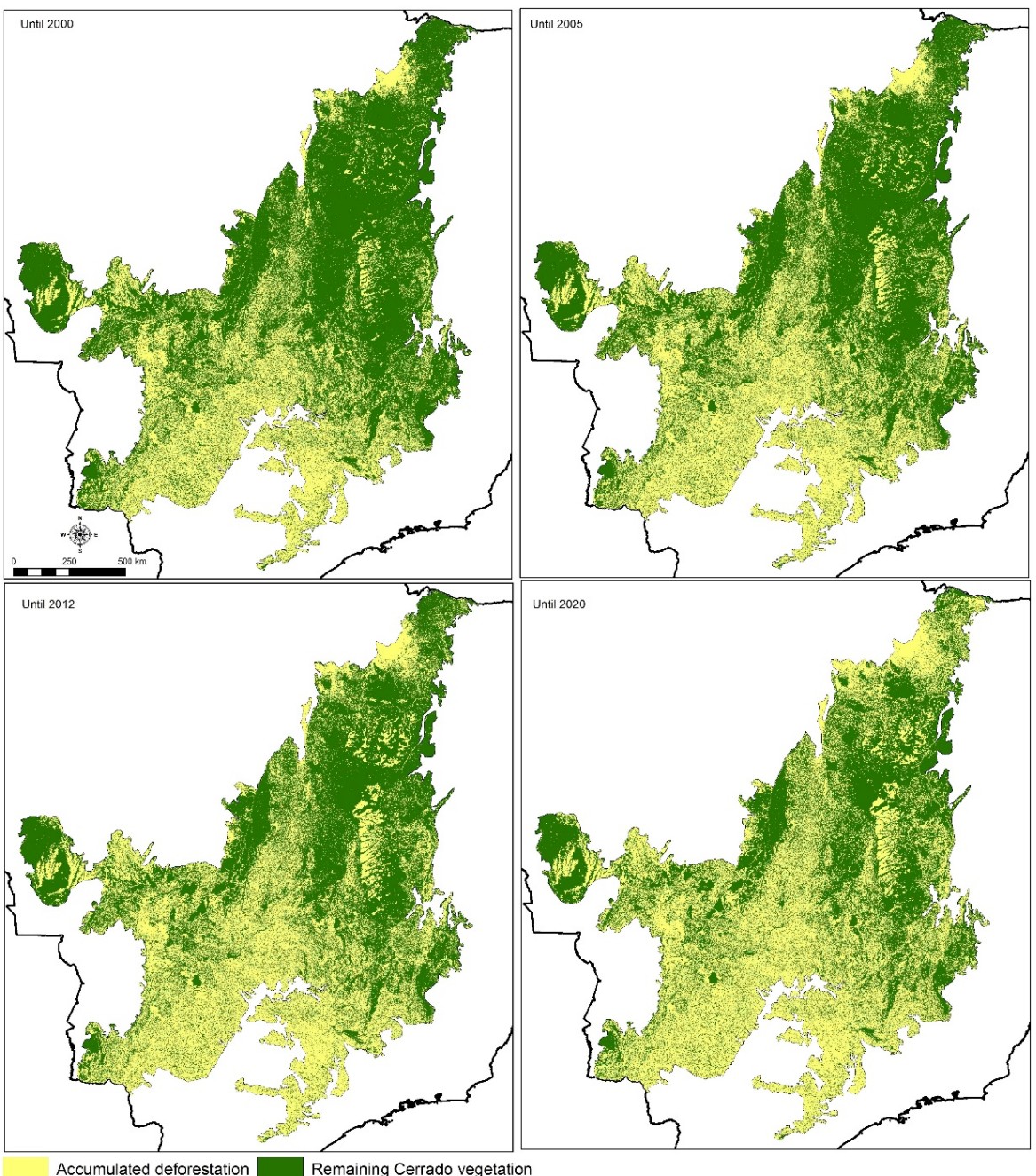

**Figure 5.** Increasing of deforestation in Cerrado 2000–2020. Source: prepared by the authors, based on [24]. Of all deforestation in Cerrado until 2020, approximately 72%, about 737,310 km$^2$ of deforested area, occurred till 2000, mainly in the southern portion. Between 2000 and 2005, the accumulated deforested area increased to 865,588 km$^2$, which represents an increase of 12.6% (128,278 km$^2$) in the deforested area. From 2006 to 2012 it was 946,867 km$^2$, increasing 7.98% (81,279 km$^2$) in accumulated deforested area. In the period from 2012 and 2020, the total accumulated deforested area was 1,018,481 km$^2$, which represents 7.03% (71,613 km$^2$) of all deforestation in the Cerrado. Note that during this period there was an expansion of deforestation in the preferential northeast direction. By the year 2000, regarding total deforestation the year 2020, there was over 72% of all biome deforestation. After five years, this percentage increased to 85%. In 2012, the accumulated total deforestation was 92.97% of total deforestation in 2020. The graph in Figure 6 illustrates the percentage of deforestation in the different periods of analysis.

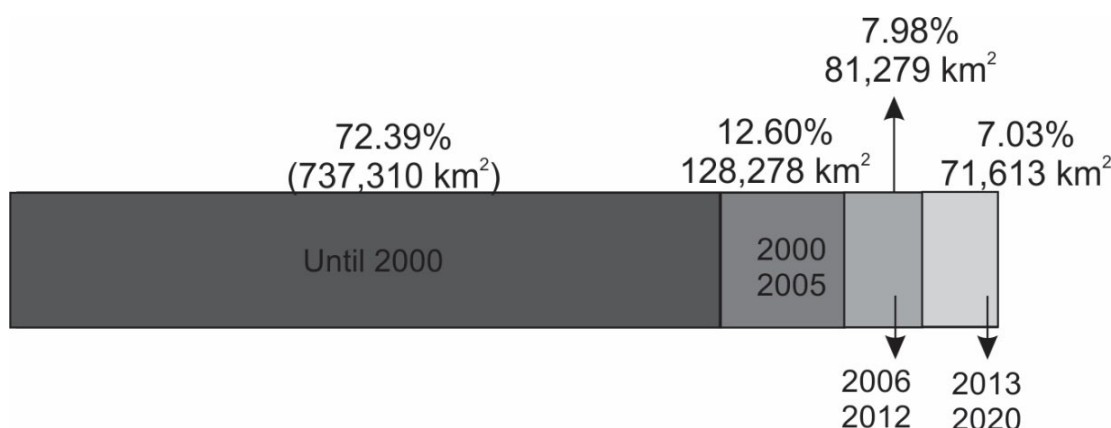

**Figure 6.** Accumulated deforestation in the Cerrado at different time intervals. Source: prepared by the authors, based on [24]. When assessing deforestation in these periods and comparing deforestation rates with the implementation of public policies it is possible to infer, that after 2005, when there was a greater presence of the state, through the creation of laws and programs to regulate actions to combat deforestation, incentives for recovery and monitoring, there were also a decrease on deforestation rates, as Figure 7. The National Program for the Conservation and Sustainable Use of the Cerrado Biome (2005), the SFB (2006), the Public Forest Management Law for Sustainable Production, the ICMBio (2007) the PNMC (2009) were created during this period.

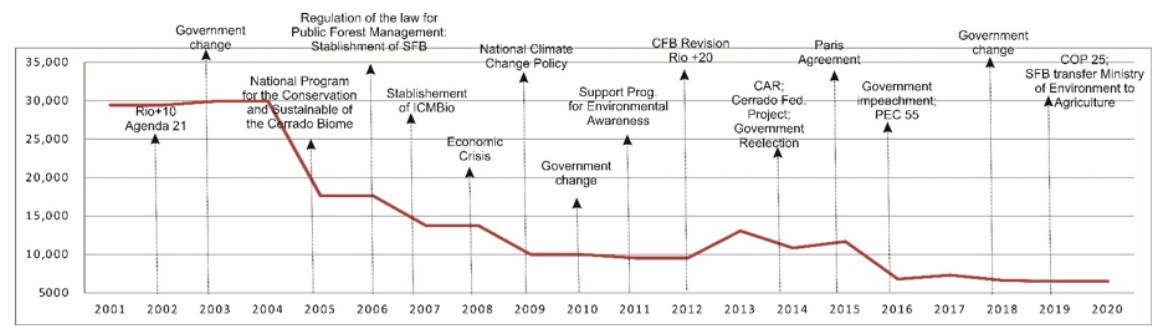

**Figure 7.** Rate of deforestation in the Cerrado 2000–2020. Source: prepared by the authors, based on [24]. After the implementation of CFB (2012), there was a general trend of a slight resumption of deforestation after a long period of decline (2004–2012). This may reflect both the less restrictive character of the new version of the law, the beginning of a growing influence of the ruralist contingent, or just a deforestation rate fluctuation. Once the law changed, the recomposition of illegal APP deforested, no longer needs to be recomposed in its entirety, but only in a width that varies according to the size of the rural property. Some landhouses (i.e., those whose size is classified as small) according to law, were exempt from the need to maintain the RL if, until 22 July 2008, they did not have enough native vegetation to compose the preservation area according to the biome. Besides, the new legislation allowed the overlap between the RL and APP areas, which was not allowed in the previous version.

## 5. The CAR and Its Contribution as a Management Tool for Controlling Deforestation

CAR is the main focus of CFB as the main instrument for management, control and monitoring of the level of adequacy of environmental legislation in rural properties in Brazil. Until December 2020, 973,061 landowners joined the CAR in the Cerrado. This number, which exceeded initial expectations, was reached after five deadline extension (Table 1).

**Table 1.** CAR deadlines and number of properties registered in Cerrado.

| Type | Date | Number of CAR | Cumulative Number of CAR | % Accumulated of CAR |
|---|---|---|---|---|
| 1st year of CAR implementation | 31 December 2014 | 74,912 | - | 7.69 |
| Initial Deadline | 6 May 2015 | 93,862 | 168,774 | 17.34 |
| 1st Extension | 5 May 2016 | 220,206 | 295,118 | 30.33 |
| 2nd Extension | 31 May 2017 | 150,308 | 445,426 | 45.78 |
| 3rd Extension | 31 December 2017 | 66,573 | 511,999 | 52.62 |
| 4th Extension | 30 May 2018 | 43,650 | 555,649 | 57.10 |
| Deadline | 31 December 2018 | 58,804 | 614,453 | 63.15 |
| Landhouse registered till | 31 December 2019 | 216,095 | 830,548 | 85.35 |
| Deadline for research data | 31 December 2020 | 142,513 | 973,061 | 100.00 |
| Overall | - | 973,061 | 973,061 | - |

Source: [19] Organized by authors.

The need for postponing deadlines is related to the landowners' level of access to information, their environmental awareness and the government itself, which has been reducing investments in the environment each year (from R$2.6 billion in 2013 to 1.9 billion in 2020, according to Comptroller General of the Union—CGU, 2020), which may have influenced the disclosure of CAR. On the other hand, one aspect that may have resulted in increased adherence to the CAR was the binding obligation of registration for granting rural credit to finance agricultural activities. With the registration the deadline on 31 December 2018, from 1 January 2019, farmers without CAR cannot access agricultural credit lines. Adherence to the CAR after the deadline does not entail any penalty for the landowner, only access to financing is prohibited, until the property is registered.

Assessing CAR implementation, in these early first years, there is a predominant trend of decreasing deforestation (as shown in Figure 8).

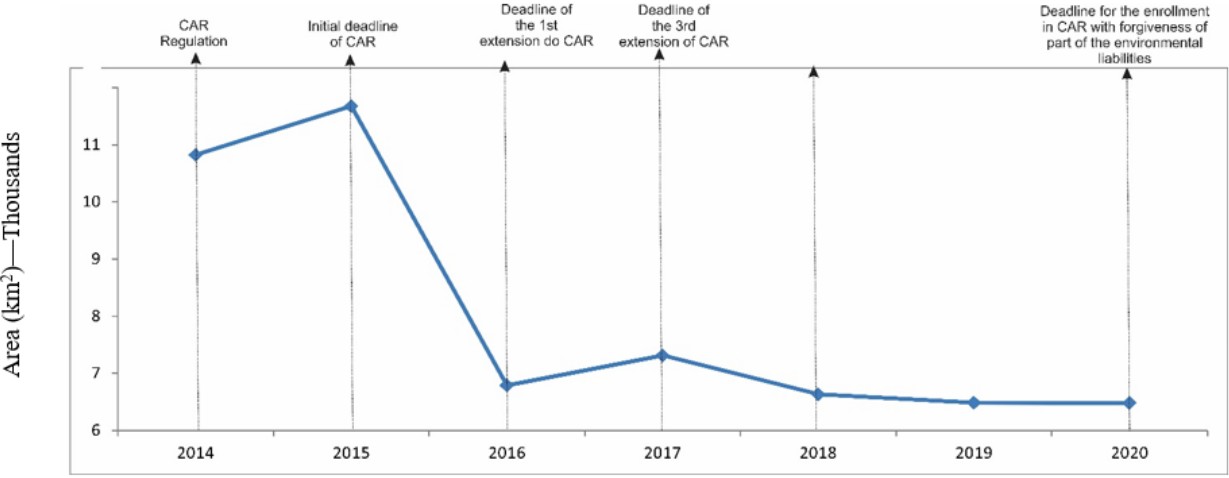

**Figure 8.** Annual deforestation after CAR regulation. Source: [19] Organized by authors.

In the first year of the CAR, there was a slight increase in the amount of deforestation, from 10,904 km$^2$ to 11,129 km$^2$. There is a hypothesis that some farmers deforested parts of their land, including it as a consolidated area, as if those areas had been deforested before 2012, hoping to be forgiven. This increase may also be related to low initial adherence. From the second year, there is a sharp drop in deforestation, of almost 17 thousand km$^2$ to 7497 km$^2$ in 2016.

The comparative analysis between the number of landhouses registered in the CAR year by year and the deforested area, shown in Figure 9, shows that there is a trend that the increase in the adhesion to the CAR is related to the decrease of deforestation in the Cerrado. Pearson's correlation coefficient between the deforested area and the number of rural properties registered in the CAR has a moderate negative correlation; r(10) = −0.81, *p* = 0.025854. Such a has already been identified in deforestation studies in the Brazilian Amazon [45,46].

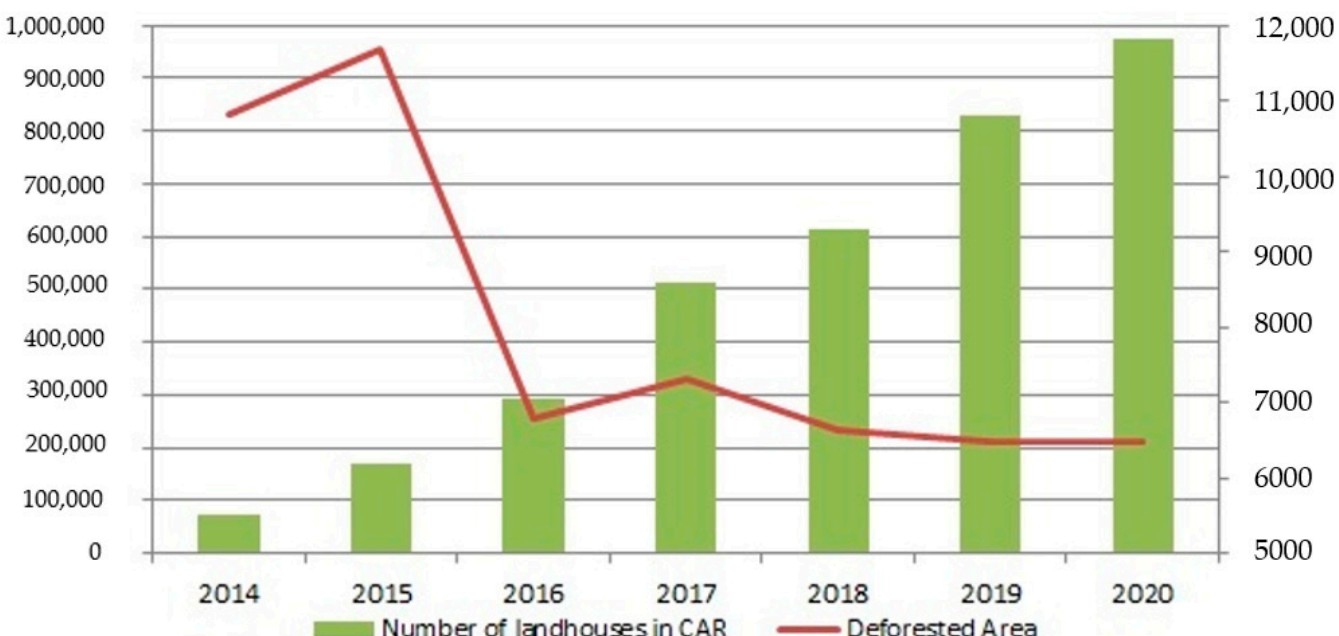

**Figure 9.** Number of landhouses in CAR x deforested area in Cerrado. Source: [19]; Organized by authors.

Although there are other variables related to the reduction of deforestation, the implementation of the CAR may be contributing to the reduction of deforestation in the Cerrado since the electronic record of the environmental regularity of the property, once analyzed by the State, may inhibit illegal deforestation. This trend should be confirmed in future evaluations, since the CAR has only been in operation for seven years.

## 6. Discussion and Conclusions

Brazil is experiencing a turbulent moment in the political scenario, which may put the goals of sustainable development and deforestation control at risk. The rise of political groups aligned with agribusiness interests may result in further changes in legislation to benefit the expansion of the sector. This may put at risk the advances experienced mainly between 2003 to 2012, when several laws were created and the current Ministry of the Environment structure was set up, who is responsible for stabilizing a period of constant decline in deforestation rates (2005–2012). After the Forest Code review in 2012, even with the amnesties granted to violators, there was a slight resumption of the increase in deforestation rates, which may be related to the rise of the influence of the agricultural sector on the policy. The approval of PEC 55/2016, which sets the spending ceiling and the investment freeze for 20 years, puts at risk the monitoring and environmental control actions. Besides, the presidential election of 2018 elected a candidate whose political basis is composed of conservative sectors, including agriculture. One of the first actions in its mandate, established through Provisional Measure No. 870, was to remove SFB, the agency that manages the CAR, from the Ministry of the Environment to the Ministry of Agriculture. This change may represent a clear conflict of interest, as ruralists will be in control of the main program responsible for the environmental recovery of rural property liabilities. This

fact has been increased environmentalists' concern, once the influence of the agricultural lobby has gained strength in recent times (illustrated, for example, by the increased release of pesticides, as showed in Figure 4).

The next steps that are part of the instruments foreseen in the Forest Code review, together with the CAR, are the Environmental Recovery Program (PRA) and the Environmental Reserve Quotes (CRA). While the first envisages actions for recovering environmental liabilities in real estate such as native vegetation suppression and improper interventions in permanent preservation areas (APP) and legal reserve (RL), the second is an environmental compensation instrument that allows producers to hold in their property a deficit of RL, acquire areas in other properties, as long as they are in the same biome, for environmental compensation.

As shown in Figure 3, protected areas in Cerrado play an important role in preventing deforestation. Once (only in 2020) inside restricted protected conservation units areas the total accumulated deforestation was 589 km$^2$ against 6483 km$^2$ in non-protected areas. However, the restricted protection areas occupy only 2.5% of the biome, which puts at risk the maintenance of the remnants of native vegetation, since the proportion of the protected area is very small—four times smaller than the number of areas protected in the Amazon, biome which has 10.5% of its territory occupied by restricting protect areas.

Through the studies developed in this work, it was possible to associate the deforestation rates with legislation changes from 2000 to 2020. As shown in Figure 5, it is possible to observe that mainly from 2005 onwards, with the creation of the Brazilian Forest Service and ICMBio, the institution responsible for the management of protected areas, there was a reduction in deforestation rates.

We divided this period into three intervals: until 2005, before the implementation of National Program for Conservation and Sustainable Use of the Cerrado Biome; a period pre-Forest Code review, until 2012; and the period after Forest Code review, after 2012 until 2020. In the first period there were 12.6% of the whole Cerrado deforestation and some legislation related to combat deforestation in Cerrado. After this period, due to the change of the political context, a tendency of resuming deforestation is ratified by 7.03% of the whole Cerrado native vegetation suppression after 2012, as illustrated in Figure 4. Especially after 2014, there was a decrease in annual deforestation rates from 1.33 to 1%, and this can be related to the growth of monitoring and implementation of CAR.

Although the program is still in its first seven years, and most of the reported information has not yet been analyzed, it was possible to speculate a relationship between the increase in property registration and the reduction of deforestation in the Cerrado from 2014 to 2020 (as shown in Figure 9).

This shows that environmental management policies such as CAR can deliver positive results even in short term. The number of information and entries in the CAR is a previously unavailable data collection for environmental management bodies. Thus, effective inspection and compliance of legislation is an increasingly accessible reality. With the establishment of CAR, which adds up to the advancement of geotechnologies and remote sensing tools, the task of monitoring and enforcing environmental compliance in a country of continental dimensions such as Brazil is becoming increasingly less difficult.

For the CAR policy to become more effective, it is necessary that the environmental agencies to advance in the analysis of the environmental regularity of properties declared in CAR. This involves greater government investment in hiring public servants and in strategies to engage farmers to comply with CAR notifications. With this, a scenario is envisioned in which the mechanisms of environmental recovery and the market for environmental reserve quotas are gaining more and more strength in Brazil, contributing to the increase of environmental regularity.

**Author Contributions:** Conceptualization, C.H.P.L.; methodology, C.H.P.L.; software, Not applicable.; validation, V.A.S.; formal analysis, C.H.P.L. and V.A.S.; investigation, C.H.P.L. and V.A.S.; resources, C.H.P.L. and V.A.S.; data curation, C.H.P.L.; writing—original draft preparation, C.H.P.L.; writing—review and editing, C.H.P.L. and V.A.S.; visualization, C.H.P.L. and V.A.S.; supervision, V.A.S.; project administration, C.H.P.L. and V.A.S.; funding acquisition, V.A.S. All authors have read and agreed to the published version of the manuscript.

**Funding:** This research received no external funding.

**Institutional Review Board Statement:** Not applicable.

**Informed Consent Statement:** Not applicable.

**Data Availability Statement:** https://drive.google.com/drive/folders/1qafiqTYT6mRyTKuCg4 IlSBjr3N4zMgV7?usp=sharing (accessed on 26 June 2022).

**Conflicts of Interest:** The authors declare no conflict of interest.

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
