# Peer review of "Recent Environmental Legislation in Brazil and the Impact on Cerrado Deforestation Rates"

_sustainability, doi:10.3390/su14138096_

Round 1
Reviewer 1 Report
The study explores the effect of Brazilian environmental policies on Cerrado deforestation rates during the period of 2001-2020. The topic is relevant to Sustainability and the article was well written. Nevertheless, I have the following comments that the authors should address.
1st Comment: The authors reported on page 8 (lines 265-268) that “Between 2000 and 2005, the accumulated deforest area became 906,978 km2, this represents a 17% increase in the total deforested area. From 2005 to 2012, it was 974,117 km2, increasing (by) 3% in accumulated deforested area.” I could obtain the 17% increase because [increase in deforestation area between 2000 and 2005/final total deforestation area = (906,978-737,310)/1,043,381] was about 17%. However, I could not obtain the 3% increase during the period 2005-2012. It is because the estimate of [(974,117-906,978)/1,043,381] gave 6.4%. Similarly, the authors stated (lines 268-269) that “In the period between 2012 and 2020, the total accumulated deforested area was 1,043,381 km2, which represents 9% of the whole Cerrado deforestation. I could not obtain the value of 9% because the calculation of [(1,043,381-974,117)/1,043,381] gave 6.6%. I wonder if the authors mistakenly typed the value at the end of 2012 as “974,117”. Please check the data again. Additionally, please check whether Figure 4 needs to be updated (or it was correct).
2nd Comment: In Table 1, the cumulative number of CAR should be calculated as the (previous) cumulative number of CAR plus the number of CAR issued between the previous deadline and the current deadline (as shown in Row 4 to the last row). Because of that, the number of CAR should be (295,118-168,774) = 126,334 for Row 3 (for the 1st Extension) but not 220,206. Additionally, the authors should note that (30.33%-17.34%) = 12.99% of the CARs were issued during the period between initial deadline and the 1st extension (i.e. ~126,000).
3rd Comment: Inconsistent information was given in Table 1 and Figure 7. For example, Table 1 and Figure 7 show that the number of CAR was 168774 in 2015, 295118 in 2016, and 511999 in 2017. However, Table 1 shows the number of CAR at the end of each year to be 614453 in 2018, 830548 in 2019, and 973061 in 2020 and these values were different from the ones shown in Figure 7. So, please update Figure 7 accordingly (based on the information shown in Table 1). Additionally, please check the correlation between the deforested area and the number of rural properties registered in the CAR (using the year-end data).
4th Comment: Please clarify the unit of “…the increased release of pesticides from 104 in 2010 to 467 in 2019” (on line 26), “…the amount of type of pesticides approved by the Ministry of Agriculture, increasing (from) 104 in 2010 to 493 in 2020” (on lines 230-231), and Figure 2. Rise in the amount of pesticides 2010-2019 on Page 7 (as the figure contained information for 2020, its title should be “…pesticides 2010-2020”). Please show clearly whether the use of pesticides was characterized by “the number of times of pesticides were released…”, “the number of types of pesticides approved…”, or “tonnes of pesticides were released…” As a matter of consistency, the data shown in the Abstract (on Line 26) should be updated to “…in 2020”.
5th Comment: In section 2, the authors stated (lines 111-116) that “In the Sustainable Protected areas there are fewer restrictions… Consequently, the amount of deforestation (13,550 km2 until 2019) inside that area is 23 times bigger than in restricted use areas (589 km2)…” while the authors stated (lines 375-377; Discussion and Conclusions) that “As shown in Figure 1, protected area… once inside restricted protected areas the total accumulated deforestation is 589 km2 against 6,485 km2 only in 2019 in no protected area…” So, please check the area of deforestation in no protected area in 2019. Should it be 13,350 km2 or 6,485 km2? Please clarify.
6th Comment: There are some typing and grammatical errors such as
- Line 8: “…recente…” should be “…recent…”
- Line 14: “…, period responsible for…” should be “…, the period responsible for…”
- Lines 54 and 68: “…doesn’t…” should be written as “…does not…” [Do not use short forms in academic articles]
- Line 57: “(RORIZ et al, 2017)” should be “(Roriz et al., 2015)”.
- Line 60: “Stefanes et al (2016)” should be “Stefanes et al. (2018)”.
- Line 60: “Santiago et al (2017)” should be “Santiago et al. (2018)”.
- Line 78: “…to be used as a governance tool, once its database concentrates…” should be written as “…to be used as a governance tool because its database concentrates…”
- Line 88: “(PACHECO, et al. 2017)” should be “(Pacheco et al., 2017)”
- Line 112: “(MITTERMEIER, 2005)” should be “(Mittermeier et al., 2005)”
- Line 124: “…didn’t…” should be written as “…did not…”
- Line 152: “…Cerrado, once this law has…” should be “…Cerrado. This law has…”
- Line 194: “…, where it most be…” should be “…, where it must be…”
- Line 249: “…deforestation data, once it has the largest…” should be “…deforestation data, since it has the largest…”
- Line 255: “2012 e 2019” should be “2012, and 2019”.
- Line 262: “INPE, 2020” should be “INPE, 2021”. [Please check it carefully because it contained the 2020 data].
- Line 263: “Figure 3. Increasing … Cerrado 2000-2019” should be “Figure 3. Increasing … Cerrado 2000-2020”. [Please change its title as this Figure contained the 2020 data]
- Line 305: “…until december 2020…” should be “…until December 2020…”
- Lines 321-322: “…in Error! Reference source not found” should be “…in Figure 6”.
7th Comment: Please add the year of publication to the following references.
- Line 424: The year of publication for Costa et al.’s article should be “2018”.
- Line 427: The year of publication for INPE…’s publication should be “2021”.
- Line 431: The year of publication for Kroger’s article should be “2017” (not 2016).
- Line 458: Please update the month and year that the authors accessed to the data from SFB. It should happen in 2021.
- Line 465: The year of publication for Stern’s article should be “2000”.
8th Comment: The following references were cited in text (but not listed in References). Please update the list of references.
- Line 37: IBGE, 2004
- Line 116: INPE/PRODES, 2020
- Line 118: ICMBIO, 2019
- Line 129: MMA, 2019
- Line 172: Law 12.512 [other Decree-Laws were provided in References].
- Line 206: CFB, 2012
- Line 234: Ministry of Agriculture (2019)
- Line 248: ASISS et al., 2019
9th Comment: The following two references were listed in References but they were not cited in the text. Please consider to delete them from the list of References:
- MMA - MINISTÉRIO DO MEIO AMBIENTE. Mapeamento do uso e cobertura do Cerrado: Projeto TerraClass Cerrado. SFB. Brasília: 438 MMA, 2015. 439
- WAPNER, P. The Changing Nature of Nature: Environmental Politics in the Anthropocene. Global Environmental Politics, v. 14, n. 469 4, p. 36-54, 2014.
Author Response
Dear Reviewer 1,
I appreciate all the notes and suggestions to improve the work. I accepted all the revisions and it is possible to follow all the modifications made with Track Changes in the attached file.

Reviewer 2 Report
The authors have well described the importance of implementing tight regulations to overcome the deforestation problem in Brazil. However, some clarifications are needed as appears in my comments, for a better quality of the manuscript.

Author Response
Dear Reviewer,
Thank you for the suggestions. Find attached the answers.

Reviewer 3 Report
This paper studies the impacts of the recent environmental legislation on Cerrado deforestation rates. The topic is meaningful. The manuscript is not well organized, and it is more like a report than an academic article. I find the article is hard to read and understand, the applied data and method are not introduced in an independent section as a regular research article does. So I don’t know how the authors get their conclusions. The section introduction did not introduce the meaning and current research status of the research topic. There are two sections numbered using the same number 2. In the first section 2, in which the authors introduce the trajectory of environmental legislation in Brazil, I think a figure or a table to summarize the environmental legislation in different years is needed. Due to the missing introduction of the applied method, it is hard to understand how the authors get their conclusions, i.e., how to prove the contribution of CAR against deforestation.
Author Response
.

Reviewer 4 Report
The theme is important, but he paper does not follow the methodology for a scientific article.
From my point of view, this paper looks like an informative article, not like a scientific paper.
Author Response
.

Reviewer 5 Report
Dear Authors,
I have reviewed the paper titled: “Recent environmental legislation in Brazil and the impact on Cerrado deforestation rates". In my opinion, the aims of the paper are germane with “Sustainability” journal topic, however, in the present form, the paper fits only in part with the international scientific standards. The paper is written with an average English level. The contribution of this paper to the scientific knowledge is acceptable but some important flaws are present in the text. I understand the difficult work done, but as a reviewer it is my duty to highlight the gaps in order to improve the research approach and its presentation to the international scientific community. Please I suggest revising the paper following the suggestions and comments reported below:
- the manuscript is interesting, and my main technical concern only regards the not correct references formatting.
- However, I am not sure if it can be presented as a scientific article being missing the basic components of a scientific paper, for instance there are no research hypothesis and discussion section is mostly a comment to the results rather than a comparison to current literature in the topic.
- I understand that this is related to the particular topic of the manuscript that forces it ti look more similar to a report than to a scientific manuscript, but I suggest to the authors to make some corrections necessary to a better understanding of this work.
Author Response
The authors' reply could be seen in the attachment.

Round 2
Reviewer 3 Report
Although the authors have made some revisions, the manuscript is still hard to understand. Here are some comments that I think the authors should address.
1. Are there any related research that can be introduced in the Introduction to demonstrate the research status of this topic, such as the impacts of environmental legislation on deforestation rates in other countries or regions and the other research on the Cerrado deforestation rates.
2. I still insist that a brief introduction of all the used data is needed in the section of Data and Methodology. Actually, there are many different data used in this study, and the authors have explained the data sources in each figure and table. It would be better the gather them together and introduce the data sources of all the used data and add some brief introduction for them.
3. The trajectory of environmental legislation in Brazil is introduced in section 3, however, it is quite hard to get the key information on the evolution of environmental legislation. It is quite necessary to summarize the key information of different environmental legislations using a timeline in a table or figure.
4. The Abstract is too long, please simplify it to highlight the main conclusions.
5. In seems that the difference between deforestation annual average rate in different periods is not obvious, so how to convince the impacts of the environmental legislation.
6. Please check the semicolon in Line 22.
Author Response
Dear Reviewer,
In anticipation of acknowledgments, find attached the response to your comments. The new version of the article already incorporates the suggestions pointed out by yours evaluation. Thanks

Reviewer 4 Report
The paper looks better, has improvements in structure, bibliography and is closer to the requirements of the journal, although I do not think it meets the requirements of a scientific article.
I recommend that all abbreviations present in the figures be explained in footnotes for a better understanding.
Author Response
In anticipation of acknowledgments, find attached the response to your comments. The new version of the article already incorporated the suggestions pointed out by yours evaluation. Thanks

Reviewer 5 Report
I have reviewed the new version of the paper titled: “Recent environmental legislation in Brazil and the impact on Cerrado deforestation rates". In my opinion, the aims of the paper are germane with “Sustainability” journal topic, however, also in the present form, the paper fits only in part with the international scientific standards. The paper is written with an average English level. The contribution of this paper to the scientific knowledge could be considered acceptable but some important flaws are present in the text and not corrected after the previous revision process. I understand the difficult work done, but as a reviewer it is my duty to highlight the gaps in order to improve the research approach and its presentation to the international scientific community. I suggest to:
· Format the citation in the text and the references style following the Sustainability template (downloadable online);
· I suggest making a further effort in order to permit to this interesting paper to reach the standard for a scientific paper, for instance there are no research hypothesis and discussion section is mostly a comment to the results rather than a comparison to current literature in the topic, please note that other works in this topic exist and the authors presented the papers following a correct scientific approach. I could understand that this is related to the particular topic of the manuscript that forces it to look more similar to a report than to a scientific manuscript, but I suggest to the authors to make some corrections necessary to a better understanding of this work.
Author Response

(The authors gave the same response as above.)

Round 3
Reviewer 3 Report
The authors have addressed most of my concerns, so I do not have more comments.
Reviewer 5 Report
Dear Authors,
I have reviewed the new version of the paper titled: “Recent environmental legislation in Brazil and the impact on Cerrado deforestation rates". In my opinion, the aims of the paper are germane with “Sustainability” journal topic, in the present form, the paper fits with the international scientific standards. The paper is written with an average English level. The contribution of this paper to the scientific knowledge can be considered acceptable.